# Intermittent Fasting: Socio-Economic Profile of Spanish Citizens Who Practice It and the Influence of This Dietary Pattern on the Health and Lifestyle Habits of the Population

**DOI:** 10.3390/nu16132028

**Published:** 2024-06-26

**Authors:** Elena Sandri, Daniele Borghesi, Eva Cantín Larumbe, Germán Cerdá Olmedo, María Jesús Vega-Bello, Vicente Bernalte Martí

**Affiliations:** 1Faculty of Medicine and Health Sciences, Catholic University of Valencia San Vicente Mártir, c/Quevedo, 2, 46001 Valencia, Spain; elena.sandri@ucv.es (E.S.); german.cerda@ucv.es (G.C.O.); 2Doctoral School, Catholic University of Valencia San Vicente Mártir, c/Quevedo 2, 46001 Valencia, Spain; 3Master’s Degree in Data Science and Business Informatics, University of Pisa, Lungarno Pacinotti 43, 56126 Pisa, Italy; d.borghesi@studenti.unipi.it; 4Degree in Data Science, Polytechnical University of Valencia, Camí de Vera, s/n, 46022 Valencia, Spain; evacantinlarumbe@gmail.com; 5Predepartmental Nursing Unit, Faculty of Health Sciences, Jaume I University, Avda. Sos Baynat, s/n, 12071 Castellón, Spain; bernalte@uji.es

**Keywords:** intermittent fasting, Mediterranean diet, healthy lifestyle, survey, Spain

## Abstract

Intermittent fasting (IF) is a dietary approach that has gained popularity in recent years. More and more Spanish people are following this eating pattern, which consists of alternating periods of fasting with periods of food intake. Its benefits include improved metabolic and vascular health and weight loss. Objectives: 1. To study the prevalence of IF among the Spanish population. 2. To explore how demographic factors influence the choice to adopt this dietary approach. Methods: A descriptive cross-sectional study was conducted on the Spanish population using the NutSo-HH questionnaire, which was constructed, validated, and disseminated by the research team through a non-probabilistic snowball sampling approach, collecting socio-demographic data and nutritional, social and lifestyle habits of the population. Results and conclusions: A valid sample of 22,181 people participated, of whom 4.59% (*n* = 1018) said they practiced IF. The data show that more middle-aged men than women practice IF. In addition, individuals who follow IF methods are less likely to have no control over their food intake, are less scared to gain weight and have a higher body image, but no differences were found related to unhealthy food and nights out. There were also no significant differences in terms of the level of education, income, size of municipality, or region of residence. In conclusion, a person who practices IF seems to have adopted a healthier lifestyle and social habits.

## 1. Introduction

Intermittent fasting (IF) is an eating pattern that alternates between fasting and eating periods [1]. There are several approaches to IF; for example, the two most common ones are alternating fasting days with non-fasting days and daily fasting by eating in a specific time frame. The first one is based on changes between days with no food consumption for 24 h and non-fasting days with normal food intake; for example, with the 5:2 method, where individuals fast for 24 h twice a week and consume a very-low-calorie diet for the remaining 5 days. Fasting can occur on consecutive or non-consecutive days. The second method is time-restricted eating, where the time during which food can be eaten is limited. Typically, eating is allowed during a 6 h period per day, with breakfast in the morning and lunch before 3 p.m. This allows for a fasting period of 14–18 h per day [2,3].

Fasting is not a new practice; it has always been around in different communities and for various reasons [4], including cultural and religious reasons [5]. However, in recent years, the IF diet has become a scientifically studied dietary pattern followed by millions of people worldwide.

The benefits of IF include weight loss [6] due to limiting the eating time, so that the opportunity to consume food is reduced, which can help control caloric intake. It is also known to improve insulin sensitivity and blood sugar regulation [7]. By alternating between periods of fasting and eating, blood insulin levels can be reduced, which can help prevent insulin resistance and type 2 diabetes. Moreover, studies have shown that it promotes cardiovascular health by lowering cardiovascular risk factors such as blood pressure, cholesterol levels and markers of inflammation [8]. Practicing IF has also been shown to have positive effects on brain function, including improving neuronal plasticity, protecting against oxidative stress and reducing the risk of some neurodegenerative diseases [9]. Some animal research suggests that intermittent fasting may be associated with increased longevity and reduced risk of age-related diseases [10]. Lastly, not only can time-restricted fasting induce autophagy and provide similar health benefits to a low-calorie diet, but it is also a suitable alternative for elite athletes who require a high caloric intake. Integrating periods of fasting can optimize metabolic health, improve recovery and maintain high performance without compromising the required energy intake [11].

Despite having several potential health benefits, intermittent fasting can also have negative side effects. Some of the most common are hypoglycemia [12], dizziness and weakness [13], which can be particularly serious if combined with the use of medication [12]. Fasting without an adequate protein replacement can also cause muscle wasting [14]. This is why IF is not suitable for everyone and it is particularly contraindicated for sensitive population groups [15], such as pregnant or breastfeeding women, people with eating disorders [16] or the elderly [17].

It has been demonstrated that socio-demographic factors play an important role in the prevalence and adoption of different dietary patterns [18,19,20]. While there are numerous studies focused on investigating the benefits of IF, there are currently just a few studies specifically investigating the socio-demographic determinants that guide the uptake of this type of dietary practice and none of them focuses on the Spanish population. A recent study explored the benefits, side effects, quality of life and knowledge of the Saudi population practicing IF and related those to socio-demographic variables of the population studied [21]. Another recent study, again conducted on the Saudi population, aimed to describe the practice of IF outside of Ramadan [22]. A third study, conducted on the Swedish population, aimed to determine the socio-demographic factors associated with adherence to a specific diet and attempts to lose weight. However, this study did not focus specifically on IF [23]. There are other studies in the literature that explore socio-demographic factors and intermittent fasting, but they focus on specific population groups; to be more specific, on the pediatric population [24], on the elderly population [25], on people with eating disorders [26] or on people with mental health disorders or pathologies [27].

Further knowledge about the prevalence of IF in the population and its socio-demographic determinants is certainly of scientific and, above all, public health interest. Understanding how the practice of IF varies according to factors such as gender, age, educational level, socio-economic status or area of residence can help to identify health disparities and develop targeted interventions for vulnerable groups. This can contribute to reducing health inequalities and promoting equitable access to beneficial health practices. Also, as this dietary practice has been associated with a range of health benefits and disease prevention [6,7,8,9,10], understanding the prevalence of the dietary practice and its determinants can help to identify population groups at risk of diet-related health problems and develop appropriate interventions. Finally, data on the prevalence and determinants of IF can influence the development of public health policies and the creation of programs aimed at promoting healthy lifestyles and improving the health of the general population. This could include educational initiatives, awareness campaigns and community intervention programs.

Also, it is well known that the habits a person develops throughout their life have a significant impact on their health [28,29,30]. When someone adopts a specific diet, such as IF, their habits can undergo significant changes [31]. It is crucial to understand in detail how these changes occur to determine both the potential benefits and the adverse effects that this dietary practice may have on a person’s health.

### Study Aim and Scope

For all the above reasons, this study focuses on analyzing a large sample of Spanish citizens living in Spain with the following objectives:Analyze the prevalence of intermittent fasting in the Spanish population,Explore the socio-demographic variables that influence the practice of this type of dietary style,Investigate the relationship between the influence of intermittent fasting on the nutritional, social and lifestyle habits adopted by the population and their impact on health.

## 2. Materials and Methods

### 2.1. Type of Study and Sampling

A cross-sectional investigation was conducted to examine the adult population of Spain, focusing on individuals aged 18 and above. Those with chronic illnesses or temporary conditions that might affect their diet (such as hospitalization, a stay in a penitentiary or living in a community) were excluded from the study.

### 2.2. Ethical Approval

Ethical guidelines set forth in the Declaration of Helsinki [32] were followed, and approval was obtained from the Research Ethics Committee of the Catholic University of Valencia (approval code UCV/2019-2020/152, date of approval, 18 June 2020). Prior to their involvement, all the participants provided informed consent.

### 2.3. Instrument

The data collection tool used in this research was a questionnaire explicitly created for this study, encompassing several sections. In the first part, participants provided anthropometric data such as weight, height, and socio-demographic variables. Subsequent sections delved into the frequency of consumption of various food groups, drinking habits, and sports and lifestyle behaviors associated with health.

The questionnaire, named the NutSo-HH Scale, was developed and validated according to rigorous methodological standards and psychometric tests were carried out [33]. To ensure the questionnaire’s validity and reliability, the validation procedures involved several steps. First, a pilot group of 53 individuals with characteristics akin to the studied population provided feedback on the questionnaire. Second, a nominal group consisting of seven health experts, including a nutritionist, two family doctors, two psychologists, a social educator, and a communication expert, participated in the validation process.

Finally, to assess the construct validity, Confirmatory Factor Analysis (CFA) was conducted using the weighted least squares robust method (WLS-MV). The model fit was evaluated using several indices: the Chi-square test (χ^2^), Root Mean Square Error of Approximation (RMSEA), Comparative Fit Index (CFI), Tucker and Lewis Index (TLI), Standardized Root Mean Square Index of Approximation (SRMSEA), and Standardized Root Mean Square Residual (SRMR). A good fit was indicated by RMSEA values ≤ 0.05, CFI and TLI values ≥ 0.95, SRMSEA values ≤ 0.95, and SRMR values ≤ 0.08.

### 2.4. Data Collection

The questionnaire was hosted on Google Forms and a non-probabilistic snowball sampling method [34] was used to distribute it, mainly through social networks. The leading platform was Instagram, where the @elretonutricional account was created, and the survey was disseminated to a varied audience with the help of various professionals, influencers, and supporters. In addition, the researchers used their personal networks, LinkedIn, Twitter, WhatsApp and Facebook, to promote the dissemination.

In addition to virtual dissemination, a significant effort was made to physically disseminate the questionnaire to ensure that it also reached those respondents who were less familiar with and active on social networks. This was conducted in collaboration with establishments throughout Spain, selected for their diverse clientele (e.g., pharmacies or tobacconists). These establishments were told about the research project, and those who wanted to join were provided with posters to display on their premises. By scanning the QR code on the poster, customers could access the questionnaire and fill it out if they wished. Data collection covered the period from August 2020 to November 2021.

### 2.5. Variables

The questionnaire covers a broad spectrum of socio-demographic and anthropometric health factors, including gender, age, birthplace, residence, occupation, education, income, weight, height, self-perceived health, diet-affecting conditions, and signs of eating disorders. Furthermore, it explores dietary patterns, frequency of food consumption, sedentary behaviors, participation in physical activity, and health-related social habits like sleep patterns, smoking, and alcohol consumption.

Most variables are qualitative, offering respondents multiple options to select. However, some variables are quantitatively continuous, including self-reported age, weight, height, and the number of minutes spent on sports per week. Additionally, discrete quantitative variables, such as self-reported health levels, follow the Likert scale format.

### 2.6. Categorization of Variables

In order to analyze the data, the responses collected for the different variables were categorized, following the criteria explained below.

For the age variable, three groups were selected: young (18–30 years), adult (31–50 years), and middle-aged and elderly (≥51 years). For the level of income, two categories were distinguished: low income (<EUR 2200/month per household) and medium–high income (≥EUR 2200/month). For the level of education, two categories were established: basic education (comprising the responses of no education, primary, secondary, high school, and vocational training) and higher education (bachelor, master and doctorate).

The nutritional variables and health habit variables were categorized on a Likert scale ranging from 1 to 4 points (as shown in Table 1), except the symptoms of eating disorders, which utilized a scale ranging from 1 to 6. This categorization offers a structured evaluation of these health-related behaviors, enabling a detailed understanding of the participants’ habits in these specific domains.

### 2.7. Healthy Eating Index for the Spanish Population (IASE)

The results of the food frequency variables were utilized to calculate the IASE (Healthy Eating Index for the Spanish population). Based on a condensed version validated by Norte and Ortiz [35], this index includes variables such as fruits, vegetables, meat, dairy, cereals, pulses, and soft drinks. It assesses the frequency of consumption of foods recommended for daily, weekly, and occasional intake, as well as dietary variety, a vital aspect of a healthy diet. Behaviors aligning with the recommendations of the Spanish Society of Community Nutrition (SENC) [36] received a score of 10, with the maximum index score being 73. Based on the IASE score obtained, the population’s nutritional habits can be categorized into three groups: “Healthy” (58.4 < IASE < 73), “Needs changes” (36.5 < IASE < 58.4), and “Unhealthy” (IASE < 36.5).

### 2.8. Data Analysis

Socio-demographic categorical variables are shown as absolute values and percentages. Continuous variables are presented as the mean and standard deviation. For the analysis of the continuous variables, the nonparametric Mann–Whitney test was used when comparing two groups or the Kruskal–Wallis test with Hochberg correction when comparing three groups [37]. A *p*-value < 0.05 was considered statistically significant. Data were analyzed using Python version 3.9.13.

Principal Component Analysis (PCA) [38] was performed in R Studio version 4.3.0 [39]. PCA is an unsupervised machine-learning model that reduces the dimensionality of a dataset by transforming a large set of variables into a smaller set of principal components, which still preserves most of the data’s variance. These principal components are new variables constructed as linear combinations or mixtures of the original variables. PCA enables the identification of the variables that best characterize Spanish IF followers and elucidates the relationships between the variables under study.

## 3. Results

A valid sample of *n* = 22,181 Spanish citizens was recruited by disseminating the questionnaire. Of the respondents, 80.84% were women compared to 19.16% men, and the average age of the participants was 34.86 years. Detailed data can be found in Table 2.

Of all the respondents, 79.22% (*n* = 17,573) claimed not to follow any diet, the other 16.19% (*n* = 3590) were divided between different dietary styles (vegetarian, vegan, raw, Weight Watchers, ketogenic, paleo, and others). Moreover, 4.59% (*n* = 1018) claimed to practice intermittent fasting.

The socio-demographic characteristics of the group practicing IF are shown in detail in Table 3. Out of this group, 235 respondents were men (23.08%) compared to 783 women (76.92%); the average age of those who practice IF was 36.78 years.

Focusing on the intermittent fasters (*n* = 1018), a Principal Component Analysis (PCA) was conducted to reduce the dimensionality of the variables. The selection of the number of proper principal components is supported by the scree plot (Figure 1), which represents the percentage of explained variance alongside the number of dimensions and by the diagrams in Figure 2a,b which indicate the contribution of the variables in each of the dimensions. It is essential to comment that the dashed red line is the average of the explained variance of all the dimensions. In this study, we chose two components, explaining the 23.7% of the data variance, following the elbow method [40].

The variables that contribute the most to the PCA model are Obesophobia, No control and Body Image, for the first dimension (a), and Fried food and Fast food for the second dimension (b). This can also be observed in Figure 3, where these variables have a more reddish color than the rest of the variables.

It is important to state that No control, Obesophobia, Body Image and BMI are highly correlated (the arrows are at 0°). Moreover, the first three variables contribute highly to the first dimension (the ones next to the horizontal line), which indicates that they capture most of the variance of the dataset. Therefore, the differences between individuals following IF are found in No control, Obesophobia, and Body Image. On the other hand, we could not observe any correlation between these variables and Fried food, Fast food or Night outings because the arrows are at a 90° angle. However, in our data, these three variables are positively correlated, indicating that respondents who usually consume fried and fast food are prone to go out at night. Finally, it was found that No control, Obesophobia, Body Image and BMI are negatively correlated with Sleep quality and Self-perceived health (the arrows are at a 180° angle). In other words, the participants with obesophobia, no control over their food intake or with body image concerns have worse self-perceived health or sleep quality than the ones who do not have these types of feelings. 

If we represent the individuals who practice IF in the two dimensions of the PCA and color them by the different socio-demographic variables, we obtain the plots shown in Figure 4, Figure 5 and Figure 6. Figure 4 shows the distribution of individuals divided into three age groups: young people (18–30 years old) in green, middle-aged adults (31–50 years old) in red and adults over 50 years old in blue.

In Figure 4, the ellipse representing young adults is displaced to the upper right side compared to the rest of the age groups. By overlapping Figure 3 and Figure 4, we can determine that young adults tend to consume more fast, fried, and ultra-processed food and that they tend go out at night more frequently. Furthermore, as the blue ellipse (>51 years) presents less displacement on the lower right side, older adults might have fewer body image concerns and obesophobia and more control over their food intake. 

This also can be observed in Table 4, which represents the health and lifestyle habits of the IF practitioners divided into the three age groups. Statistically significant differences can be seen in the frequency of weekly consumption of fried food between young and middle-aged people (2.06 vs. 1.93; *p* = 0.03) and between young and old people (2.06 vs. 1.78; *p* < 0.001). The same trend is observed for the consumption of fast food (Y = 2.28 vs. M = 2.18 and L = 1.74; *p*-value Y-L: <0.001; *p*-value M-L: <0.001), for the consumption of ultra-processed products (Y = 2.25 vs. M = 2.03 and L = 1.70; *p*-value Y-M: <0.001; *p*-value Y-L: <0.001; *p*-value M-L: <0.001) and for nightlife (Y = 1.28 vs. M = 1.09 and L = 1.07; *p*-value Y-L: <0.001; *p*-value M-L: <0.001).

Figure 5 shows the sample divided into men and women and colored accordingly. It is observed that the ellipse created by the answers of the masculine contestants (in red) is displaced to the upper left side, where Self-perceived health and Sleep quality are found. Therefore, we can conclude that men assigned themselves a higher self-perceived health punctuation and a better sleep quality than women. On the other hand, the ellipse created by the answers of the feminine contestants (in green) is displaced to the lower right side, where Obesophobia, No control, and Body Image are located (Figure 3). In a nutshell, females tend to have more concerns about their body image, gaining weight, and not having control over their food intake than men.

In Table 5, these results are expressed numerically, and the described relationships are confirmed by calculating the *p*-values using the Mann–Whitney test for the indicated variables.

Figure 6 (a) shows the distribution of people following IF according to the size of the municipality of residence (blue < 2000 inhabitants; green 2000–10,000 inhabitants; red > 10,000 inhabitants), (b) according to the level of education (green: basic level of education; red: higher education), (c) by income level (green: low income; red: upper-middle income; gray: don’t know–no answer), (d) by whether a person lives with (red) or without family (green) and (e) by whether the respondent lives alone (red) or not (green). The data indicate that none of these socio-demographic variables show any statistically significant differences in terms of the analyzed social and lifestyle habits.

Finally, Figure 7 shows the percentage of people practicing IF in each autonomous Spanish community. To find out between which regions statistically significant differences occur, Dunn’s test was carried out, the results of which can be found in detail in Figure A1 of Appendix A. It was found that the practice of IF was highest in the region of Murcia and lowest in La Rioja.

## 4. Discussion

In our study, we explore lifestyle habits as well as health perceptions among a sample of Spanish adults following an IF diet to better understand the practice of this dietary approach. IF, as a highly intriguing clinical nutritional approach, is more and more recognized in our society not only for its health benefits [22,41,42,43] but also for the convenience of practicing it [44,45]. A growing popularity has been noticed as well as an increasing interest among the Spanish population, decreasing the interest in the Mediterranean diet [46]. Therefore, IF is currently postulated as a public health tool to prevent diseases and to improve quality of life.

First, it is essential to highlight the high level of people’s engagement observed in this study. Participants exhibited remarkable interest toward the central topic, “nutritional and social habits”. Of particular note is the predominance of female participants, who provide 80.84% (*n* = 17,930) of the data of the sample. This fact leads to an imbalance between the number of men and women who reported following an IF diet, with 783 women (76.92%) out of the total 1018 IF practitioners in our sample. This gender distribution may reflect either a heightened interest among women in diet interventions [17] or that men are less likely to participate in surveys [22,42,47]. An exception is presented in a study conducted on the Saudi population, which showed a slightly higher participation of men compared to women [21].

The results obtained from the PCA indicate that individuals who feel they lack control over the amount of food they consume, experience guilt, show fear of gaining weight, or who are concerned about their body shape are more likely to engage in intermittent fasting. Various studies have demonstrated that the concepts of guilt or control related to food impact consumers’ eating behavior and, therefore, the choice of a specific diet and lifestyle [48,49]. Perceptions of guilt have the potential to lead consumers to seek healthier and more sustainable food solutions by positively influencing their food choices, eating behavior, purchasing decisions, and lifestyle [50]. Several authors have investigated the personal, social, and/or cultural motivations behind a consumer’s choice of one diet over another; for example, in Finland [51], Australia [52], Germany [53] or in the USA [54], concluding that understanding the effect of control or guilt on food choices and eating behaviors could be significant in promoting healthy eating behaviors.

Furthermore, it has been observed that individuals experiencing obesophobia also exhibit concerns regarding their body image and opt for intermittent fasting as a dietary approach. Recent studies, such as the one carried out in Australia by Sutin et al. [55], indicate that people with a fear of gaining weight may experience weight-based discrimination, which can lead to unhealthy eating behaviors or dietary choices [56,57,58]. In this last case, IF is not advisable and would be contraindicated [16].

Regarding body image and its relationship with IF, a greater concern about body image is associated with a higher likelihood of engaging in IF behaviors. To explain this relationship, it is necessary to consider that body image is a complex and subjective construct that incorporates perceptions of size, satisfaction with appearance, attractiveness evaluation, and body control practices [59,60]. Available research conducted on a sample of Australian men and women suggests that a more tolerant and positive attitude toward body shape and size is associated with a lower likelihood of engaging in intermittent fasting behaviors [61], and individuals display less concern about weight changes and adopt positive health behaviors, as we see in studies with respondents from a variety of countries [62,63] or in one from with participants from Brazil [64]. The findings of other studies from the USA [65,66] or Spain [67] also indicate that concern about body weight control significantly influences a person’s perception of their own body, as evidenced in this study.

Another important finding of the PCA is that fried and fast food consumption is not related to the behavioral variables Control, Obesophobia and Body Image, and vice versa. This means, for example, that the increase or decrease in fast food consumption of an IF practitioner does not provide information about the fear of gaining weight. Integrating fried or fast food into an IF diet is not advisable due to them typically containing high amounts of calories, fats, and refined sugars [68]. Regular consumption of these types of food can contribute to a variety of negative health outcomes, including cardiovascular diseases, type 2 diabetes, increased risk of obesity, and certain types of cancer, which is backed up by studies conducted both worldwide [69] and focusing on just one nation, such as Spain [70], Colombia [71] or the USA [72].

Hence, there is a need to address not only the nutritional implications but also the broader health consequences associated with these types of alimentation, making it necessary to identify the best strategy to reduce any negative impact their consumption may have on public health [73].

One final observation drawn from the PCA is that participants with a higher body weight report poorer self-perceived health and worse sleep quality than those with a normal weight. Our data indicate that 31.43% (*n* = 320) of the population practicing IF is overweight or obese, which implies, in addition to poorer self-perceived health, a greater fear of gaining weight [56] as well as poorer sleep quality, with reduced sleep hours, insomnia, sleep apnea, and daytime sleepiness. Evidence on this topic can be found in Brazil [74], the USA [75], Japan [76] and Greece [77]. However, as indicated in a Dutch clinical trial, IF can improve sleep through interfering with circadian rhythms [78] or by reducing body weight, as shown in some studies carried out in the USA [79,80,81].

Regarding age, it is known that younger individuals have a lower BMI than older adults, a trend also observed in studies from other countries; for example, in the USA [82,83] or Norway [84]. However, losing and maintaining a healthy weight is a primary concern for people of all age groups [42], as evidenced in our study by observing that the mean BMI in all the age groups is lower than 25, which indicates a healthy weight. On the other hand, older adults report lower consumption of fried food, fast food, and ultra-processed food than young adults or middle-aged adults. Age is a factor that can influence food choices [85,86], but it is not the only one. There is a great body of evidence indicating that food preferences change throughout life due to various factors such as sensory characteristics (texture, taste, smell) and perceptual features (color, size, quality) of foods [87,88,89,90], social environment [91], psychological characteristics [92] physiological needs [93], habits [94], and cognitive factors (nutrition knowledge, skills, attitudes, tastes, and preferences) [95].

Young people who practice IF report better sleep quality than older adults. In a study by McStay et al. [96] in the USA, it is indicated that the effects of IF on sleep quality are uncertain because it is necessary to consider whether there is an underlying problem such as sleep apnea or insomnia. However, existing evidence from Italy indicates that aging influences in the structure and quality of sleep [97].

Lastly, regarding age, it is interesting to mention that the fear of gaining weight or the concern about body shapes affects young people who practice IF significantly more. The literature shows that these concerns can occur in any age group [98]. However, in young adults, social pressure, beauty standards, and the influence of the media can contribute to a greater concern about body image and fear of gaining weight [99,100], potentially leading to restrictive diets or excessive exercise. Conversely, in older adults, although concern about body image may still be relevant, priorities shift with ageing, with health and overall well-being becoming more important [17].

If we reflect on the health and lifestyle habits by gender of the surveyed individuals practicing IF, in our study, women have a lower BMI, consume fewer fried and fast foods, report poorer sleep quality, and also perceive their health as poorer. In contrast, men feel as if they have less control over the amount of food they consume, less fear of gaining weight, and are less concerned about their body image. According to data from the Informe Anual del Sistema Nacional de Salud 2022 [101], men (79.3%) report a better health status than women (71.9%), a pattern observed in our study of the IF practitioners, too. The same report states that obesity in Spain affects 16.0% of the population aged 18 and over (16.5% men and 15.5% women). In the existing literature, overweight or obese individuals experience more fear of gaining weight than underweight or normal weight participants [56], with women again reporting more fear of gaining weight, which indicates more concern about weight [102].

Regarding sleep problems, Spanish women report greater difficulties falling asleep and waking up multiple times at night [103]. Moreover, some studies worldwide [104,105] have found that men are significantly less dissatisfied with their bodies compared to women. Others, on the contrary, such as the study by Millstein et al. in the USA [106], inform that an important percentage of women are satisfied with their appearance and have a positive judgement of excess weight.

In this study, the size of the municipality of residence, the level of education, the income level, if the individual is living with family or not and if the individual is living alone or with other people were not found to have a significant association with the habits followed by those who practice IF. However, we discovered the following observations in other papers. In rural areas with smaller populations, there are indications of certain problems with optimal dietary intake or reports of barriers to accessing certain foods [107]. Individuals with higher income levels can access healthier food and maintain better nutrition [108]. They also have a lower BMI, according to the Encuesta Nacional de Salud de España [109]. This same survey also indicates that a higher level of education is associated with a lower BMI, implying greater awareness of the importance of a balanced diet and the potential health risks associated with incorrect or inadequate nutrition [108]. Moreover, it has been observed that living alone or in an arrangement without family connections influences some nutritional patterns and health habits [110], with individuals living with family having a higher BMI than those living elsewhere [111].

Regarding the percentage of people practicing intermittent fasting for each autonomous community in Spain, the results seem to indicate that there is no clear pattern. A higher adherence to this type of diet is observed in the region of Murcia and, on the contrary, a lower use of the diet is observed among the population of La Rioja, but this difference is not statically significant, it just marks a slight trend. Nevertheless, a possible explanation for this, as indicated in the Informe del Consumo Alimentario en España 2021 [112], is that in the northern regions of the country such as the northwest and north center, meals are more often consumed outside of the individual’s own living space compared to the national average, while the Levante region has a higher number of meals consumed at home. The report also indicates that the region of Murcia exceeds the national average of the per capita consumption of meals outside of home by the end of 2021. Conversely, La Rioja is the autonomous community with the lowest per capita consumption. In La Rioja, more olive oil, wine, meat, fish, milk and derivatives, and bread are consumed than in Murcia. On the other hand, in Murcia, more rice (with consumption higher than the national average), cereals, fresh fruits and vegetables, eggs, and legumes are consumed. The more frequent consumption of olive oil, wine, meat, fish, milk and derivatives, as well as bread, could be influenced by the local gastronomic tradition in La Rioja, as well as by the natural resources available there, such as olive groves and vineyards.

In contrast, the greater consumption of rice, cereals, fresh fruits and vegetables, eggs, and legumes in Murcia suggests a more varied diet in terms of agricultural products. Murcia is known for its intensive agriculture and its climate benefits the cultivation of a wide variety of fruits, vegetables, and cereals, which may influence the dietary preferences of the population. All these presented statements about regional food consumption may not only be influenced by local preferences but also by the climatic conditions and the availability of products in each region. Therefore, we can see how cultural, geographical, and economic factors might influence the dietary habits of a community.

Statistically significant differences in the IF practitioners are observed when comparing the regions of Valencian Community and Catalonia, Valencian Community and Murcia, Castile/León and Murcia, and Galicia and Murcia. According to the latest data published by the Instituto Nacional de Estadística in 2024 [113], the highest percentage of young adults and middle-aged adults in these autonomous communities is found in the region of Murcia, with 18.88% and 36.96% respectively. In contrast, the lowest percentage of the older adult population is also found in the Region de Murcia, with 44.16%. These official data are consistent with the results obtained in this study, where we can observe that those communities with a higher proportion of young individuals show a greater interest in the IF model.

### Strengths and Limitations

The main strength of this study is that we could verify the influence of lifestyle and health habits on the population practicing the IF model, which can aid in the creation of public health programs that support and promote the controlled use of this type of diet to ensure the physical and psychological well-being of the population, avoiding harmful eating habits and sedentary lifestyles. Another undeniable strength of this study is the extensive sample size, encompassing diverse demographics in terms of age, education, and income levels. Furthermore, geographical representation across all the regions of Spain ensures a comprehensive understanding of the health status, as well as nutritional and lifestyle habits, of the entire Spanish population. This breadth of diversity enhances the reliability and robustness of the study’s findings.

One limitation of the study is that we investigated people who practice IF and the relationship with lifestyle and health habits, but we did not study the type of IF or duration and frequency of IF. Future research could explore these dimensions, including both the length and regularity of IF behaviors, to achieve a more comprehensive understanding of this phenomenon.

Another limitation to highlight is that the data collection was based on self-reported data to evaluate the association between IF and lifestyle and health habits. Consequently, these data cannot be independently verified, so we cannot totally exclude biases such as selective memory, exaggeration, or attribution. Future research should consider the use of objective measurement tools to validate the research results, thus promoting the accuracy of the results.

The last weakness of this study is attributed to the sampling method employed, primarily utilizing snowball sampling and prevalent dissemination through social media. While snowball sampling facilitates reaching a wide array of responses, it carries the risk of self-selection bias, as the sample composition depends on initial participants recommending additional participants. In social networks, influencers play a significant role in disseminating content related to specific topics, attracting followers who often share similar interests and behaviors. It is plausible that the survey reached some of these communities through influencers, potentially also influencing respondents’ behaviors and preferences. To minimize this bias, a major effort has been made to also disseminate the questionnaire outside the social networks, as described above.

## 5. Conclusions

The intermittent fasting diet offers a wide array of health benefits for various conditions, like obesity, diabetes mellitus, cardiovascular diseases, cancer, and neurological disorders. However, it is essential to note that this dietary approach is not suitable for all individuals. It is necessary to individualize it to obtain better health results without suffering unnecessary risks.

Therefore, it is important to understand how intermittent fasting relates to other determinants such as a person’s lifestyle and their socio-demographic variables.

In the present study, the PCA highlights that the main variables in which the individuals following an intermittent diet are different from each other are Lack of Control, Obesophobia, and Body Image. Additionally, respondents with obesophobia, lack of control over food intake, or concerns about body image reported worse self-perceived health and sleep quality compared to those without such concerns.

Regarding age, younger individuals tend to consume more fast food, fried foods, and ultra-processed foods, and they go out at night more frequently. In contrast, older adults seem to have fewer body image concerns, less obesophobia, and more control over their food intake.

As for gender, men reported a higher self-perceived health scores and better sleep quality than women. Women tend to have more concerns about their body image, weight gain, and lack of control over food intake.

Socio-demographic variables such as the size of the municipality of residence, education level, income level, living alone or with others, and living with family or independently did not show statistically significant differences concerning the habits followed.

Finally, regarding the geographic region of residence, it was found that the practice of intermittent fasting was more common in the region of Murcia and less common in La Rioja.

The results obtained are a starting point for understanding the interaction between intermittent fasting, lifestyle habits and socio-demographic variables in the Spanish population. Although this information needs to be further expanded and extrapolated to other populations, this understanding may help to design more effective public health and preventive strategies.

## Figures and Tables

**Figure 1 nutrients-16-02028-f001:**
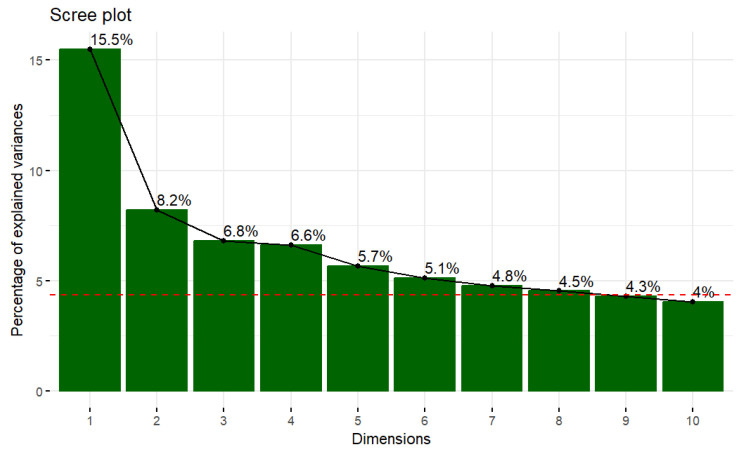
Scree plot of the PCA model.

**Figure 2 nutrients-16-02028-f002:**
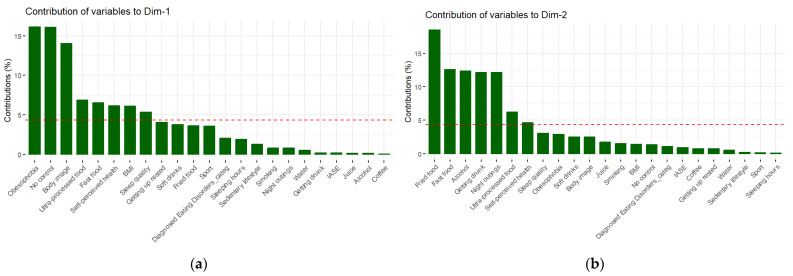
(**a**) Contribution of the variables to the first dimension of the PCA model. Note: The red line indicates the average percentage of contribution if all the dimensions contribute the same. (**b**) Contribution of the variables to the second dimension of the PCA model. Note: The red line indicates the average percentage of contribution if all the dimensions contribute the same.

**Figure 3 nutrients-16-02028-f003:**
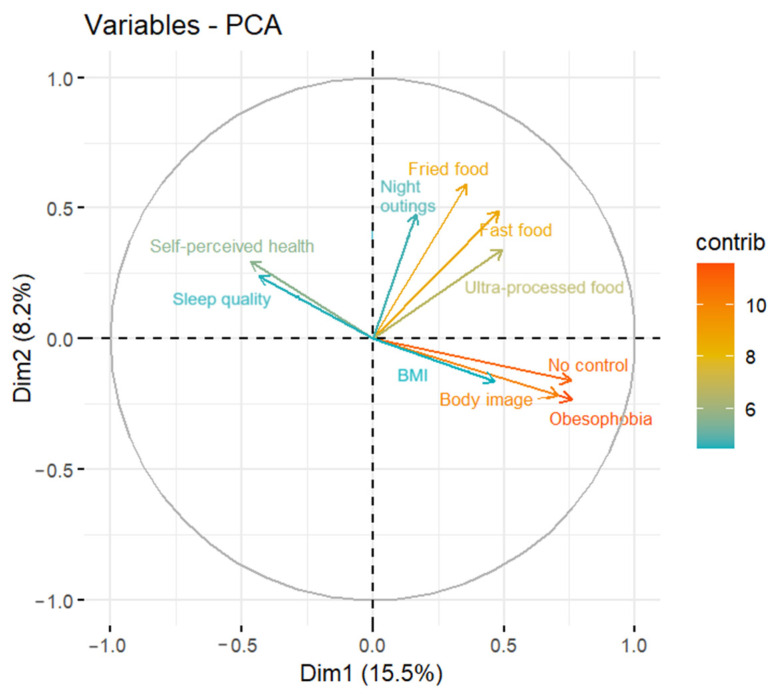
Variables plot of the PCA model. (Note: only the top ten variables contributing the most to the PCA model are represented.).

**Figure 4 nutrients-16-02028-f004:**
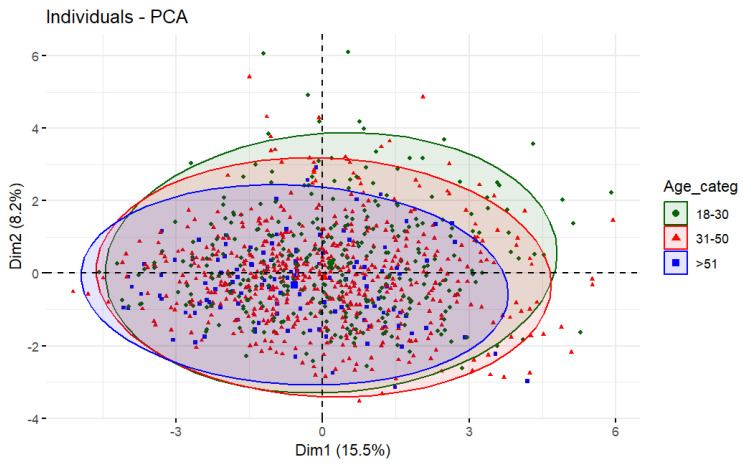
Individuals plot of the PCA model depending on age.

**Figure 5 nutrients-16-02028-f005:**
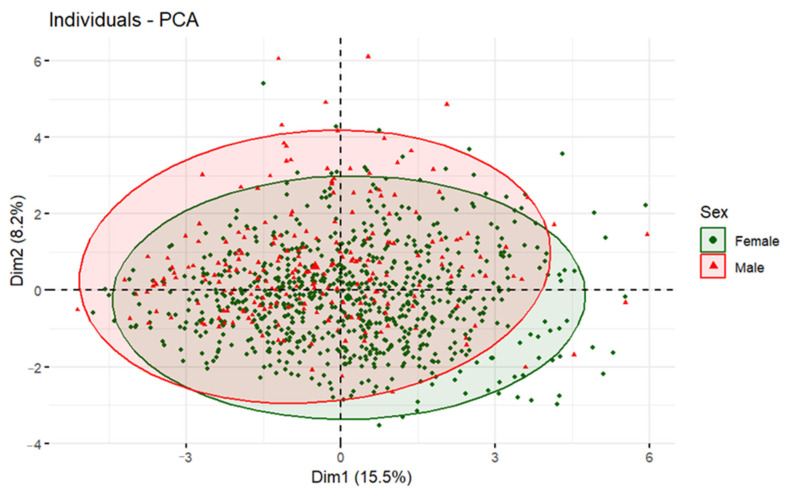
Individuals plot of the PCA model depending on sex.

**Figure 6 nutrients-16-02028-f006:**
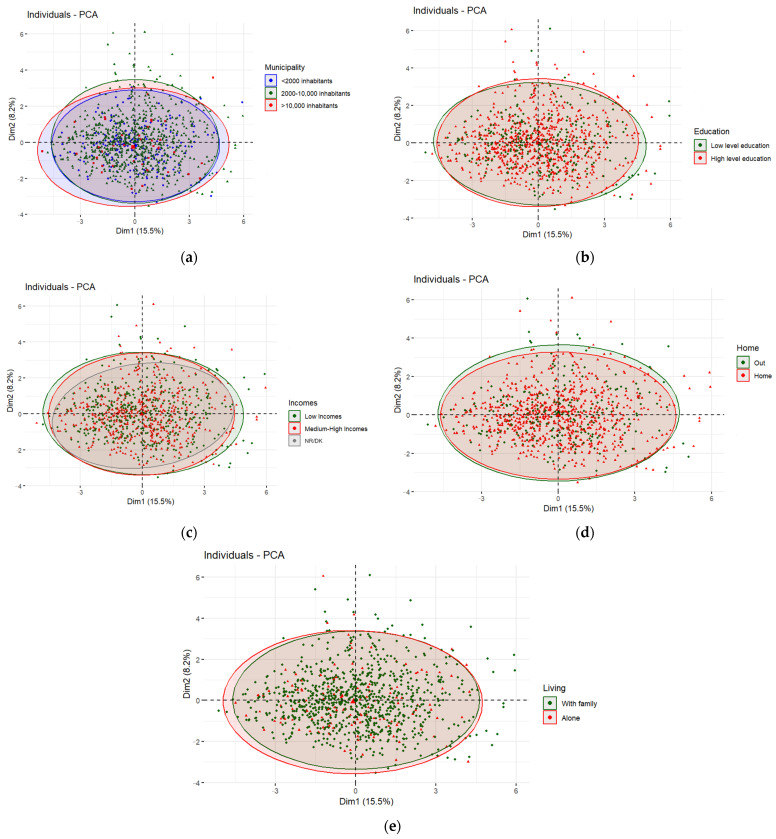
(**a**) Individuals plot of the PCA model depending on municipality. (**b**) Individuals plot of the PCA model depending on education. (**c**) Individuals plot of the PCA model depending on incomes. (**d**) Individuals plot of the PCA model depending on home. (**e**) Individuals plot of the PCA model depending on living.

**Figure 7 nutrients-16-02028-f007:**
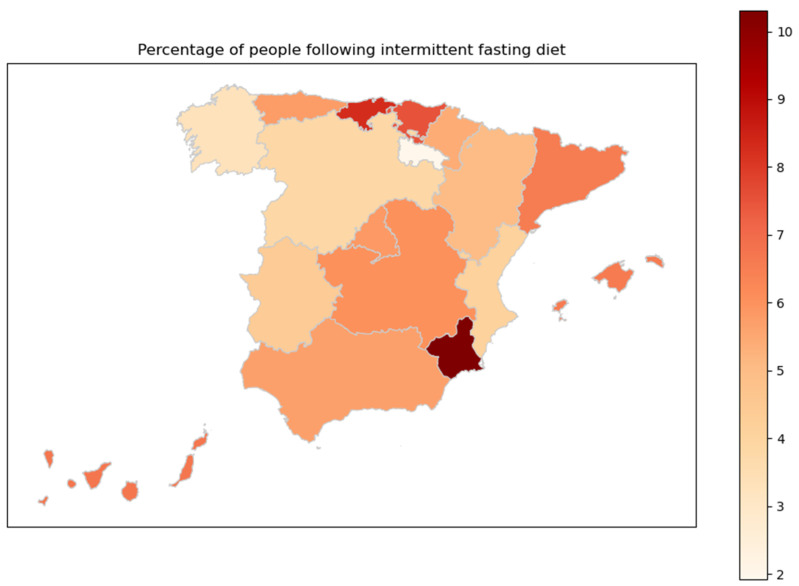
Percentage of people practicing intermittent fasting for each autonomous community in Spain.

**Table 1 nutrients-16-02028-t001:** Categorization of the health and lifestyle variables.

Variable	Category	Score
Sleeping hours	<6 h	1
	6 h–7 h	2
	7 h–8 h	3
	>8 h	4
Getting up rested	Never	1
	Very seldom and sometimes	2
	Frequently and almost always	3
	Always	4
Sleep quality	0 and 1	1
	2	2
	3	3
	4 and 5	4
Water	Never and very rarely (2 max. per month) or 1–2 glasses/cups per week	1
	2 glasses/cups or less every day	2
	3 to 5 glasses every day	3
	More than 5 glasses every day	4
Sugary soft drinks, coffee and energy drinks, juice	Never and very rarely (2 glasses max. per month)	4
	One glass per week and 2 or more glasses per week	3
	2 glasses or less every day	2
	3 to 5 glasses and more than 5 glasses every day	1
Consumption of fast food, fried food and ultra-processed dishes	Never	1
	Very seldom (2 times a month maximum)	2
	Once a week	3
	Several times a week	4
Getting drunk	Never or less than once a month	1
	Monthly	2
	Weekly	3
	Daily or almost daily	4
Alcohol consumption	Never or once a month	1
	2–4 times a month	2
	2–3 times a week	3
	4–5 times a week or every day	4
Smoking	Non-smoker	1
	Light smoker (less than 5 cigarettes per day)	2
	Moderate smoker (6–15 cigarettes per day)	3
	Severe smoker (more than 16 cigarettes per day)	4
Night outings	Never and sporadically	1
	Between a and 2 night a week	2
	More than 3 times a week	3
	Every day	4
Daily sedentary lifestyle	Less than 7 h	1
	Between 7 and 9 h	2
	Between 9 and 11 h	3
	More than 11 h	4
Obesophobia, no control, body image	Always	6
	Very frequently	5
	Frequently	4
	Occasionally	3
	Rarely	2
	Never	1

**Table 2 nutrients-16-02028-t002:** Sample and their socio-demographic characteristics (*n* = 22,181).

	Mean ± SD or *n* (%)
Male	4251 (19.16%)
Female	17,930 (80.84%)
Age (years)	34.86 ± 11.70
Male Age (years)	34.47 ± 11.22
Female Age (years)	36.50 ± 13.41
	Total
Education level	
Basic education	15,154 (68.32%)
Higher education	7027 (31.68%)
Income level	
Low	9727 (43.85%)
Medium–high	10,616 (47.86%)
Don’t know–no answer	1838 (8.29%)
Municipality	
<2000	1014 (4.57%)
2000–10,000	3587 (16.17%)
>10,000	17,580 (79.26%)

**Table 3 nutrients-16-02028-t003:** Sample and their socio-demographic characteristics (*n* = 1018).

	Mean ± SD or *n* (%)
Men	235 (23.08%)
Women	783 (76.92%)
Age (years)	36.78 ± 10.71
Age men (years)	37.10 ± 10.79
Age women (years)	35.73 ± 10.38
	Total
Education level	
Basic education	724 (71.12%)
Higher education	294 (28.88%)
Income level	
Low	510 (50.10%)
Medium–high	446 (43.81%)
Don’t know–no answer	62 (6.09%)
Municipality	
<2000	41 (4.06%)
2000–10,000	169 (16.74%)
>10,000	808 (80.00%)
BMI (Kg/m^2^) (WHO *)	
Underweight (<18.5)	11 (1.08%)
Normal range (18.5–24.9)	687 (67.49%)
Overweight (25–29.9)	218 (21.41%)
Obesity (≥30)	102 (10.02%)

* World Health Organization.

**Table 4 nutrients-16-02028-t004:** Health and lifestyle habits of the three age groups of the respondents who practice intermittent fasting.

Habits Variable	Young Adults (18–30 Years) Mean ± SD	Middle-Aged Adults (31–50 Years) Mean ± SD	Older Adults (>51 Years) Mean ± SD	*p*-Value **
BMI	23.51 ± 3.52	24.50 ± 4.37	24.29 ± 4.61	<0.001 Y-M: <0.001
Fried food	2.06 ± 0.76	1.93 ± 0.75	1.78 ± 0.71	<0.001 Y-M: 0.03 Y-L: 0.001
Fast food	2.28 ± 0.70	2.18 ± 0.75	1.74 ± 0.64	<0.001 Y-L: <0.001 M-L: <0.001
Ultra-processed food	2.25 ± 0.87	2.03 ± 0.88	1.70 ± 0.82	<0.001 Y-M: <0.001 Y-L: <0.001 M-L: <0.001
Self-perceived health	3.91 ± 0.74	3.85 ± 0.88	3.88 ± 0.85	0.84
Sleep quality	3.54 ± 0.92	3.41 ± 0.99	3.29 ± 0.90	0.03 Y-L: 0.04
Night outings	1.28 ± 0.49	1.09 ± 0.31	1.07 ± 0.28	<0.001 Y-L: <0.001 M-L: <0.001
Obesophobia	3.79 ± 1.51	3.68 ± 1.44	3.35 ± 1.42	0.02 Y-L: 0.02
No control	2.97 ± 1.31	2.96 ± 1.25	2.65 ± 1.12	0.05
Body image	4.01 ± 1.33	3.77 ± 1.31	3.54 ± 1.26	<0.001 Y-M: 0.02 Y-L: 0.001

** Kruskal–Wallis test with Hochberg correction.

**Table 5 nutrients-16-02028-t005:** Health and lifestyle habits by sex of the respondents who practice intermittent fasting.

Numerical Variable	Woman Mean ± SD	Man Mean ± SD	*p*-Value ***
BMI	23.76 ± 4.19	25.41 ± 3.78	<0.001
Fried food	1.92 ± 0.75	2.06 ± 0.77	0.01
Fast food	2.12 ± 0.74	2.31 ± 0.74	<0.001
Ultra-processed food	2.07 ± 0.89	2.03 ± 0.87	0.52
Self-perceived health	3.80 ± 0.85	4.11 ± 0.72	<0.001
Sleep quality	3.39 ± 0.85	3.58 ± 0.72	0.004
Night outings	1.15 ± 0.39	1.16 ± 0.40	0.56
Obesophobia	3.89 ± 1.43	2.98 ± 1.38	<0.001
No control	3.07 ± 1.26	2.47 ± 1.15	<0.001
Body image	3.94 ± 1.29	3.44 ± 1.35	<0.001

*** Mann–Whitney test.

## Data Availability

The data presented in this study are available upon reasonable request from the corresponding author. The data are not publicly available due to privacy.

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
