# Peer review of "Intermittent Fasting: Socio-Economic Profile of Spanish Citizens Who Practice It and the Influence of This Dietary Pattern on the Health and Lifestyle Habits of the Population"

_nutrients, 2024, doi:10.3390/nu16132028_

Round 1

Reviewer 1 Report

Comments and Suggestions for Authors

The present study is an interesting one, but the authors should perform a series of modifications and data interpretation, in a larger populational setting, so that the results would be relevant for a broader audience except Spanish.

First off all correct the abbreviated IF that appears as AI, in Spanish, probably due to a former presentation of data in the native language, and review with attention the vocabulary and grammatical issues in the text.

The discussion section should contain parallels with similar studies conducted in other countries, so that the relevance of the conclusions is extended.

The conclusions are rather scarcely presented, so please revise carrefully this part.

Comments on the Quality of English Language

First off all correct the abbreviated IF that appears as AI, in Spanish, probably due to a former presentation of data in the native language, and review with attention the vocabulary and grammatical issues in the text.

Author Response

Please find attached in the Word document the answers. Thank you.

Reviewer 2 Report

Comments and Suggestions for Authors

As the key outcome, intermittent fasting is vaguely defined and being self-reported. There are multiple types of IF and the efficacy would differ by types, study population and other implementation methods. Without a clearly defined outcome, the study findings would be flawed. Apart from the outcome, description of other covariates being included is so brief and it is unclear whether these are validated tools. For the statistical analysis, not sure why PCA is used instead of regression analysis to examine the associated factors of IF implementation. Despite a relatively large sample size, the validity of study finding is a concern.

Comments on the Quality of English Language

Nil

Author Response

Please find attached in the word document the answers. Thank you.

Reviewer 3 Report

Comments and Suggestions for Authors

Dear Authors,

Congratulation for a well conducted study, clear results and deep discussion!

Our main concern is that your conclusions, in our opinion, are not the reflect of the main results of your study, as it is supposed to be. Please highlight the main results of your study in this part. For example:

Individuals who feel they lack control over the amount of food they consume, experience guilt, fear gaining weight, or are concerned about their body shape are more likely to engage in intermittent fasting.

Increase or decrease in fast food consumption in a consumer practicing IF does not provide information about the fear of gaining weight.

….

And finish with practical implications of your study and suggestion for future studies

Other suggestions:

In your introduction you only present the IF as a hypocaloric alternative, but does autophagy obligatory necessitate hypocaloric diet to have health effect? Time-restricted fasting (TRF), should also be an interesting alternative, especially in sports elites that need more calories? Maybe you should also discuss those points, but feel free not doing this, because it is not absolutely essential for your article.  

In instrument part, please add some references about the procedure used to construct the questionnaire.

Table 1, Water, point 1 something is lacking in the text. Sedentary lifestyle, may be Daily sedentary lifestyle is better.

Line 203, please give a reference for this procedure and specify what PSA is, like: reference for this procedure. And improve the explanation of this procedure like: Principal Component Analysis (PCA) was performed in R Studio ... (reference) to select the variable that better characterise Spanish IF ...

Line 233, please put a reference for the elbow method.

Lines 262-264 and lines 268-269, please specify that they are results from your study.

Lines 264, weig   ht

Line 399 suppress the (

Wishes of success!

Author Response

(The authors gave the same response as above.)
